# Kinetic Study and Modeling of the Degradation of Aqueous Ammonium/Ammonia Solutions by Heterogeneous Photocatalysis with TiO$_2$ in a UV-C Pilot Photoreactor

Juan C. García-Prieto [1] , Luis A. González-Burciaga [2], José B. Proal-Nájera [2] and Manuel García-Roig [1,*]

1 Centro de Investigación y Desarrollo Tecnológico del Agua, Universidad de Salamanca, Campo Charro s/n, 37080 Salamanca, Spain; jcgarcia@usal.es
2 Instituto Politécnico Nacional, CIIDIR-Unidad Durango, Calle Sigma 119, Fracc. 20 de Noviembre II, Durango 34220, Mexico; luis.gonzalez.iq@gmail.com (L.A.G.-B.); jproal@ipn.mx (J.B.P.-N.)
* Correspondence: mgr@usal.es

**Abstract:** The degradation mechanism of NH$_4^+$/NH$_3$ in aqueous solutions by heterogeneous photocatalysis (TiO$_2$/SiO$_2$) and photolysis in UV-C pilot photoreactor has been studied. Under the conditions used, NH$_4^+$/NH$_3$ can be decomposed both by photolytically and photocatalytically, without disregarding stripping processes. The greatest degradation is achieved at the highest pH studied (pH 11.0) and at higher lamp irradiation power used (25 W) with degradation performances of 44.1% (photolysis) and 59.7% (photocatalysis). The experimental kinetic data fit well with a two parallel reactions mechanism. A low affinity of ammonia for adsorption and surface reaction on the photocatalytic fiber was observed (coverage not higher than 10%), indicating a low influence of surface phenomena on the reaction rate, the homogeneous phase being predominant over the heterogeneous phase. The proposed reaction mechanism was validated, confirming that it is consistent with the photocatalytic and photolytic formation of nitrogen gas, on the one hand, and the formation of nitrate, on the other hand. At the optimal conditions, the rate constants were $k_3$ = 0.154 h$^{-1}$ for the disappearance of ammonia and $k_1$ = 3.3 ± 0.2 10$^{-5}$ h$^{-1}$ and $k_2$ = 1.54 ± 0.07 10$^{-1}$ h$^{-1}$ for the appearance of nitrate and nitrogen gas, respectively.

**Keywords:** TiO$_2$ photocatalysis; UV-C; photolysis; ammonia; nitrogen removal; water treatment

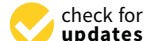



## 1. Introduction

One of the major concerns regarding environmental problems is the high concentration of nitrogen in surface and groundwater from industrial discharges, urban wastewaters and poor agricultural practices leading to over-fertilization of soils that allows the diffuse contamination of nitrogen and phosphorus, among others. According to the EEA briefing 'National Emission Ceilings (NEC) Directive reporting status 2019', emissions of ammonia from the agricultural sector continue to rise. The emissions increased by 0.4% from 2016 to 2017 with a general increase of 2.5% between 2014 and 2017, posing a challenge for EU Member States in meeting EU pollution limits. The EEA briefing notes that a more substantial reduction will be required for all pollutants if the EU is to achieve its emission reduction commitments by 2030 [1].

Ammonia emissions can lead to an increased acid deposition and excessive nutrient levels in soil, rivers or lakes, which can have a negative impact on aquatic ecosystems and cause damage to forests, crops and other vegetation. Inorganic compounds are very common in urban and industrial wastewaters, as well as nitrogen compounds, ammonium and nitrates formed from the decomposition of organic nitrogen compounds [2]. The discharge of nitrogen compounds, especially NH$_4^+$/NH$_3$, into receiving watercourses causes serious problems: in aquatic ecosystems, eutrophication, reduction of dissolved oxygen, lethality in fish [3,4] and in human health, both directly, as they attack the respiratory system, skin

and eyes, and indirectly, as they generate chloramines, carcinogenic compounds, in chlorination treatments to obtain tap water for consumption [5]. They also cause corrosion in copper pipes [6].

$NH_4^+/NH_3$ removal in industrial waters is carried out with technologies, such as chemical precipitation, electrolysis, adsorption, stripping, ultrafiltration and ion exchange [7–11]. $NH_4^+/NH_3$ removal in urban wastewaters is carried out either by biological treatments by heterotrophic bacteria under anoxic conditions and high COD/N ratio, or by autotrophic bacteria under anaerobic conditions (anammox processes) under low COD/N ratio conditions [12]. In low-cost wastewater treatment plants, nitrogen removal is carried out by natural phyto and geo-depuration systems [13]. However, these processes are costly and often ineffective. For example, the runoff water from urban WWTP sludge, on which this study is based, after dewatering treatment, contains concentrations of around 1000 mg/L of $NH_4^+/NH_3$, which cannot be discharged directly into the public watercourse and significantly increases the concentration of nitrogen to be treated in the plant, when it is recirculated to the plant headworks [14]. Traditional treatment methods are not effective enough to remove such high concentrations of $NH_4^+/NH_3$ [15].

Advanced oxidation process technologies and heterogeneous photocatalysis by $TiO_2$, in particular, have been demonstrated to be a technology capable of decomposing $NH_4^+/NH_3$ into harmless gases, such as nitrogen and hydrogen [16–20], proving to be an effective solution to this problem. $NH_4^+/NH_3$ degradation by immobilized $TiO_2$, as a photocatalyst, has been the most studied and widely used due to its abundance, non-toxicity, chemical stability, excellent UV light activity and also to avoid the limitation of separating $TiO_2$ powder from the suspension after cleaning wastewater [21]. Moreover, different immobilizations of semiconductors on solid supports have been studied [22–26]. The photocatalytic oxidation of $NH_4^+/NH_3$ in aqueous phase has been mainly investigated controlling solution pH, catalyst concentration [27–33] and intensity of irradiant light [16,34]. This advanced oxidation process involving photo-generated hydroxyl radicals as primary oxidants oxidizes $NH_4^+/NH_3$ to either nitrogen gas [35], to nitrite and nitrate anions, or both [36].

Furthermore, since 1959 it had been observed that the photolysis of ammonia occurs in the presence of nitrogen oxides to give rise to nitrogen gas [37]. Ammonia degradation by direct photolysis with hydroxyl radicals [38,39] and photo-electrochemical process [40] have been under research recently. The photochemical process based on UV sources could be a feasible technology for the treatment of $NH_4^+/NH_3^-$ containing wastewaters.

There may be a synergistic effect between the photolytic and photocatalytic processes occurring in the photoreactors to explicate the mechanism for the removal of $NH_4^+/NH_3$ from aqueous solutions. This paper presents the sensitivity analysis of the main variables and the mechanism for ammonia removal by heterogeneous photocatalysis using $TiO_2$ supported on $SiO_2$ in a pilot UV-C photoreactor, considering both photolytic and photocatalytic processes.

## 2. Results

### 2.1. Kinetic Study of Ammonia Removal

Some authors have studied ammonia degradation reactions using $TiO_2$ photocatalyst. These authors indicate that hydroxyl ions are generated from the decomposition of water, which facilitates the oxidation of ammonia to nitrogen gas or nitrate depending on the mechanism followed [14,41,42].

A basic pH benefits the reduction of ammonia/ammonium:

$$NH_4^+ + H_2O \rightarrow NH_3 + H_3O^+ \ pKa\left(NH_4^+\right) = 9.246$$

$$NH_3 + OH^\bullet \rightarrow NH_2 + H_2O + e^-$$

$$NH_2 + OH^\bullet \rightarrow NH + H_2O + e^-$$

$$N + N \rightarrow N_2$$

Some intermediate species react with each other in the presence of protons to form nitrogen gas as well:

$$NH_x + NH_y \rightarrow N_2H_{x+y} \, (x, y = 0, 1, 2) + H^+ \rightarrow \ N_2$$

On the other hand, a part of the ammonia species is oxidized to nitrite, either by the reaction of the oxygen with two holes created on the surface of the photocatalyst or with the solvent molecules [41]:

$$NH_3 + O_2 + 2\,\hbar^+ \rightarrow NO_2 + 3\,H^+$$

$$NH_3 + 2\,H_2O + 6\,\hbar^+ \rightarrow NO_2 + 7\,H^+$$

Nitrites, in turn, oxidize into nitrates:

$$NO_2 + H_2O + 2\,\hbar^+ \rightarrow NO_3 + 2\,H^+$$

Whether the formation of one product or the other is favored depends on the characteristics of the medium, the pH and the oxygen concentration, either favoring a basic pH, in the absence of oxygen the decomposition of ammoniacal nitrogen into gaseous nitrogen or in combination [43].

Serewicz and Noyes [37] observed that photolysis of ammonia occurs in the presence of nitrogen oxides to give rise to nitrogen gas, nitrous oxide and small amounts of hydrogen. Other authors [38,39] have investigated the oxidation pathway of photolytic ammonia oxidation in presence of $OH^{\bullet -}$ radical. Huang et al. [38], using the laser flash photolysis technique with transient absorption spectra of nanosecond, proposes that hydroxyl ions could oxidize $NH_3$ to form $NH_2$ as the main product of photolytic oxidation. It would further react with OH$\cdot$ radical to yield NHOH. Since NHOH could not remain stable in solution, it would rapidly convert to $NH_2O_2^-$ and consequently $NO_2^-$ and $NO_3^-$. Wang et al. [39] suggest the following reaction mechanism:

$$NH_3 + 6\,OH^{\bullet} \rightarrow NO_2 + 4\,H_2O + H^+$$

$$NO_2 + 2\,OH^{\bullet} \rightarrow NO_3 + H_2O \,\text{(rapid)}$$

$$NO_3 + H_2O + hv \rightarrow NO_2 + O_2 + 2\,H^+ \,\text{(slow)}$$

In order to study the optimal conditions for ammonia degradation, as well as the contribution of the photocatalytic and photolytic processes, a series of studies were carried out, as follows.

### 2.2. Influence of Different Factors on the Heterogenous Photocatalytic Process of $NH_4^+/NH_3$ Removal

Firstly, a sensitivity analysis of the optimal conditions for $NH_4^+/NH_3$ removal by heterogeneous photocatalysis in the UBE photoreactor was carried out. It is known that the factors that most influence $NH_4^+/NH_3$ degradation the most, in addition to temperature, are pH, conductivity, dissolved oxygen and lamp irradiation power (P).

The influence of pH is due to the fact that the photocatalytic reactions take place on the surface of the photocatalyst and they are strongly influenced by that surface charge, which is different in acidic or alkaline conditions. The $TiO_2$ isoelectric point is around 6.5 [44], then at lower solution pH its surface would be positively charged, whereas at higher pH the surface would be negatively charged [45]. Consequently, the electrostatic properties of catalysts' surfaces in different environments with respect to reactive compounds play an important role on the rate of the reaction [46]. Furthermore, the speciation of the $NH_4^+/NH_3$ acid/conjugate base pair at the different working pHs must be taken into account. pKa of ammonium is 9.25, thus, at pH 7.0, $TiO_2$ is near to neutrality and $NH_4^+$ (99.4%) has a positive charge and no electrostatic attractive forces appear, at pH 9.0 $TiO_2$ is

negatively charged and $NH_3$ (36%) / $NH_4^+$ (64%), then significant attractive forces between photocatalyst surfaces and $NH_4^+$ occur, finally at pH 11.0 $TiO_2$ is negatively charged and $NH_3$ (98.3%) /$NH_4^+$ (1.7%) and almost no attractive forces should appear. Considering that degradation of $NH_4^+$/$NH_3$ increases at higher pHs, the authors infer that attractive forces between pollutant particles and photocatalyst surface do not play an essential role in these experimental conditions. Several authors indicate that ammonia oxidation takes place only under alkaline conditions [27,29,34,47]. In this sense, others authors proposed that at basic pH, the scenario is different and the adsorption of neutral $NH_3$, rather than $NH_4^+$ on the surface of $TiO_2$, may be the rate-limiting step [28]. Thus, Shibuya et al. [32] observed at pH 10 that increasing the amount of molecular ammonia adsorbed by the catalyst is important for enhancing the oxidation rate. At the same time, at high pHs, the negative charge of the photocatalyst surface hinders the adsorption of other anions and aromatic organic compounds as they probably exist as anionic species at these pHs [48–50]. Furthermore, at higher pHs the concentration of $H^+$ significantly decreases, reducing its competition with $NH_4^+$ for exchange sites on the photocatalyst surface, thus favoring the efficiency of ammonium removal [16,27,31].

Figure 1 shows the kinetic curves of the concentrations of ammonium/ammonia, nitrate and nitrogen gas. The kinetic curve for the concentration of nitrogen gas released is calculated by the difference after the corresponding mass balance. Under these conditions the values for gas-phase species are relative values, not absolute values.

The nitrite concentration measured was always in the order of 100 times lower than the nitrate concentration, but since the rate of nitrite oxidation at lower pH is faster than at a higher pH [32], then the nitrite concentration should increase with pH. Thus, it is considered that nitrite is a very unstable intermediate species that is rapidly transformed into nitrate or nitrogen gas and therefore the nitrite concentration has not been considered, nor represented, for kinetic purposes. In this respect, Wang et al. [39] recently observed that ammonia is more reactive than ammonium ion with hydroxyl radical (OH.), by a stepwise $H_2O_2$ addition method, $NH_4^+$/$NH_3$ can be completely converted to $NO_x^-$ in UV-C photolytic process. The low formation of nitrate could be explained by pH effect and another route involves photo-reduction of nitrate to form nitrite or nitrogen gas by photolytic and photocatalytic processes [40,51–53].

Other authors [31,42] indicate that when the concentration of dissolved oxygen increases significantly, the concentration of nitrite and nitrate in the solution increases as well. In the experiments carried out in this work, no oxygen has been introduced into the solution, except for the oxygenation that may occur as a consequence of the recirculation flow rate (1000 L/h) of the sample in the photoreactor circuit. At the experimental conditions followed in the $NH_4^+$/$NH_3$ degradation kinetics, measurements of the dissolved oxygen concentration of the samples have shown that there are no noticeable changes in the dissolved oxygen concentration, as shown in Figure 2.

However, conductivity of the samples increases with time in the ammonium degradation experiments, observing that the difference between the initial conductivity (time zero) and the final conductivity (7 h) is greater with increasing pH and lamp irradiation power, which indicates that more ionic intermediate species are being formed (Figure 3).

Therefore, after this sensitivity analysis, it could be stated that the percentage of $NH_4^+$/$NH_3$ removal is mainly a function of lamp irradiation power and pH. In this sense, the results shown in Figure 4 indicate that the degree of $NH_4^+$/$NH_3$ removal increases with pH at both lamp irradiation powers. This is consistent with what has been observed by other authors for homogeneous photocatalytic processes [16,31]. Considering that pKa of ammonium is 9.25, the ammonium ion is the major species at pH 7.0 and pH 9.0, while molecular ammonia is predominant at pH 11.0.

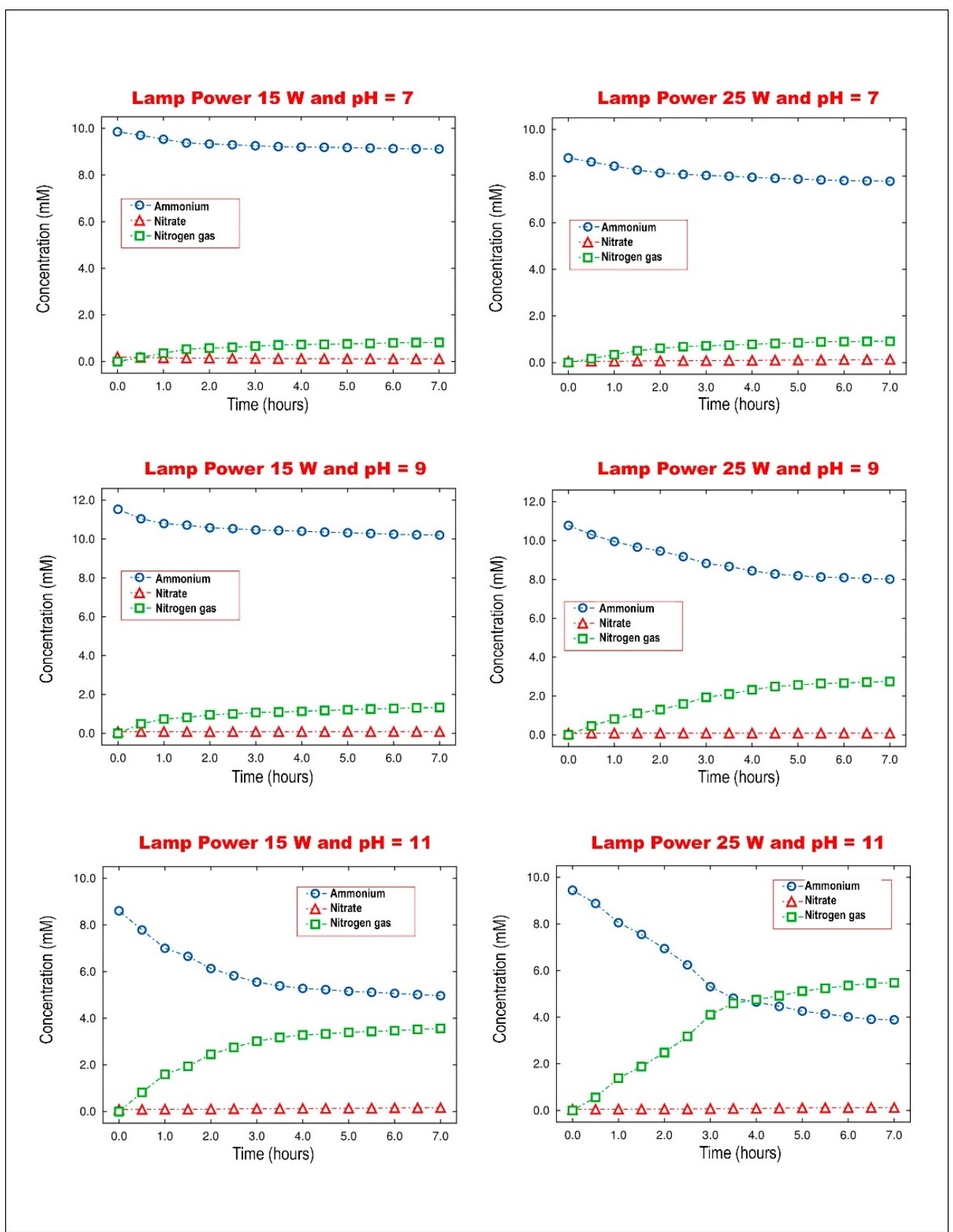

**Figure 1.** Influence of pH (pH 7.0–11.0) and lamp irradiation power (15 W, 25 W) on the performance and kinetics of $NH_4^+/NH_3$ removal by photocatalysis. Q = 1000 L/h, temperature 20 ± 1 °C.

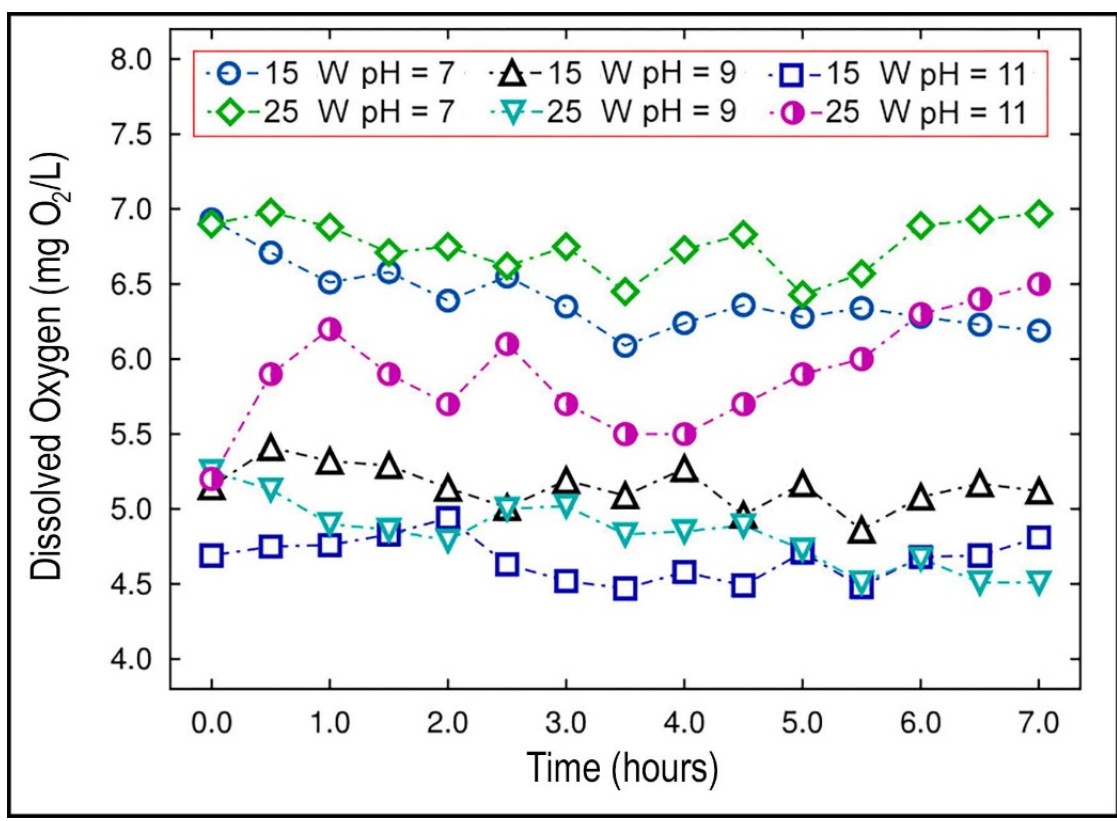

**Figure 2.** Dynamics of dissolved oxygen in the samples during $NH_4^+/NH_3$ removal process by photocatalysis. Q = 1000 L/h, temperature = 20 ± 1 °C.

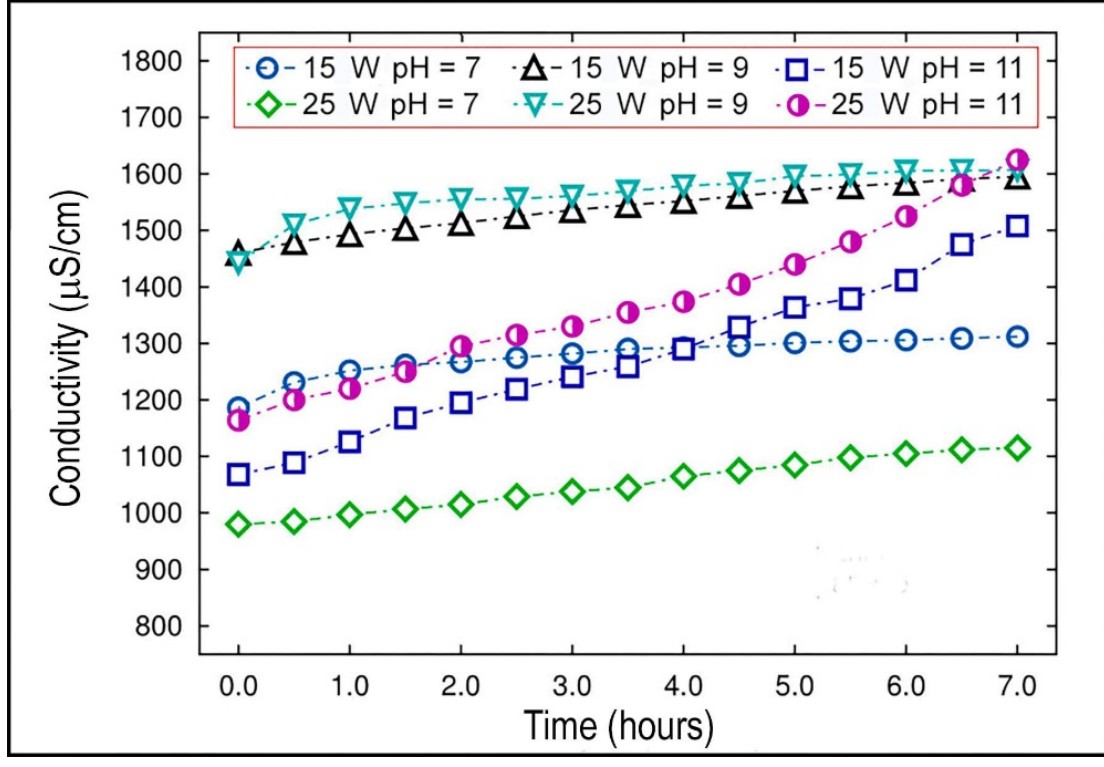

**Figure 3.** Dynamics of sample conductivity during $NH_4^+/NH_3$ removal process by photocatalysis. Q = 1000 L/h, temperature = 20 ± 1 °C.

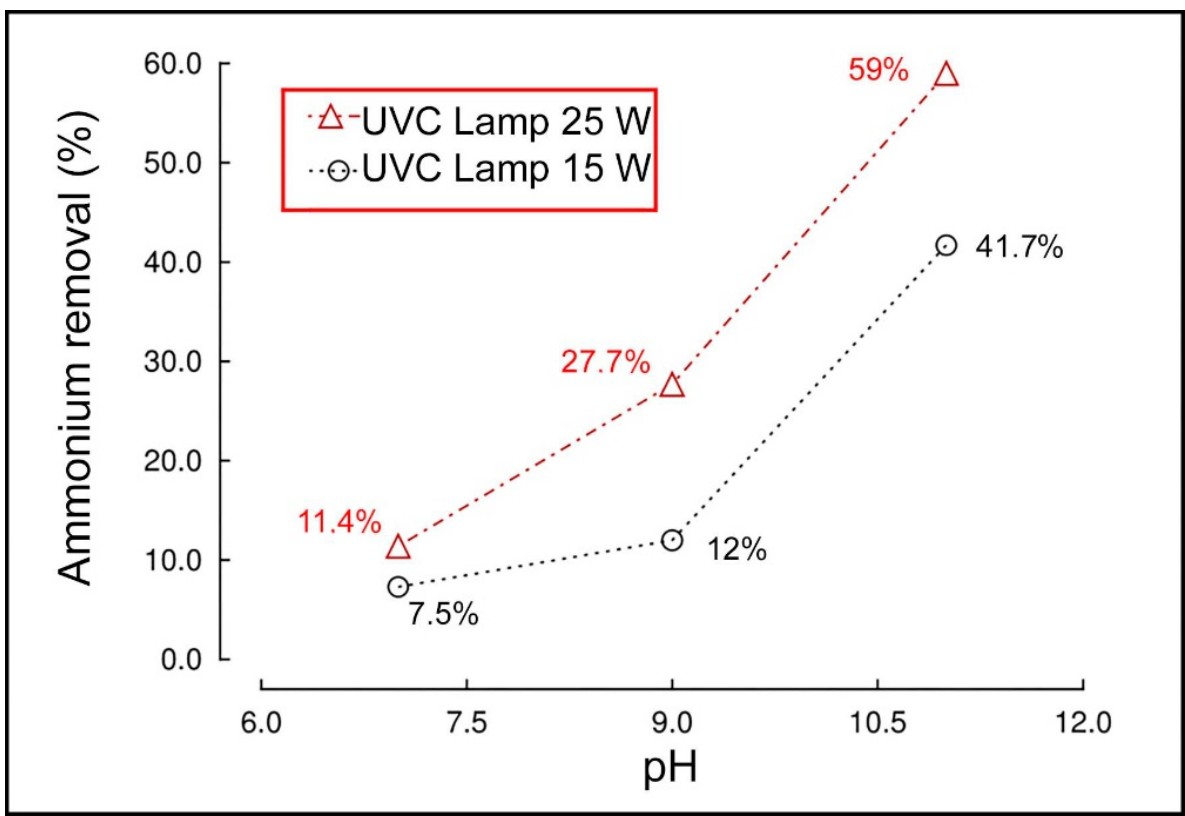

**Figure 4.** Dependence of $NH_4^+/NH_3$ removal performance (at 7 h), by photocatalysis, on pH (7.0–11.0) and lamp irradiation power (15 W, 25 W). Q = 1000 L/h, temperature = 20 $\pm$ 1 °C.

Previous studies of fiber stability against acids and bases indicates that the fiber is stable with respect to pH, except at extreme pHs, with loss of titanium at very acidic pHs and loss of silicon at very basic pHs (higher pH = 11) [54].

*2.3. Influence of Stripping, Photocatalysis and Photolysis on the Kinetics of Ammonia Removal*

According to the optimal conditions, three experiments were carried out with similar initial ammonium concentrations (180–200 mg/L), at pH = 11.0, 25 W UV-C lamp, Q = 1000 L/h and T = 20 $\pm$ 1 °C (Figure 5). This was done so in order to study the effect of ultraviolet light (photolysis effect) and the effect of stripping or degassing, as a consequence of the recirculation flow rate of the sample (Q = 1000 L/h), on the performance of ammonia removal by heterogeneous photocatalysis. This stripping will produce a decrease in the concentration of ammonium in the solution as a result of the passage, by volatilization, of the dissolved ammonium into the air. In this sense, a series of experiments were carried out as follows in order to check the different effects on the kinetics of ammonia degradation: the first experiment with the lamp on and photocatalyst (photocatalytic effect), the second experiment with the lamp off and photocatalyst (stripping effect) and the final experiment without photocatalyst but with the lamp on (photolysis effect). Figure 5 shows the percentage of degradation as a function of reaction time in each of the three experiments carried out. The highest performance is by photocatalysis (59.7 %) but a high removal by photolysis (44.1%) is also observed, which may be caused by photochemical reactions (photolysis) with nitrogen oxides formed during recirculation (stripping) with ammonia in the presence of UV-C light [37,40]. In this context, the effect of nitrogen removal by stripping (14.4%) is significant, which suggests that part of the ammonia may be lost from the solution either by volatilization, adsorption on the catalyst surface or both, not to mention the fact that ammonia may oxidize into nitrogen oxides due to agitation and aeration of the mixture during recirculation of the solution.

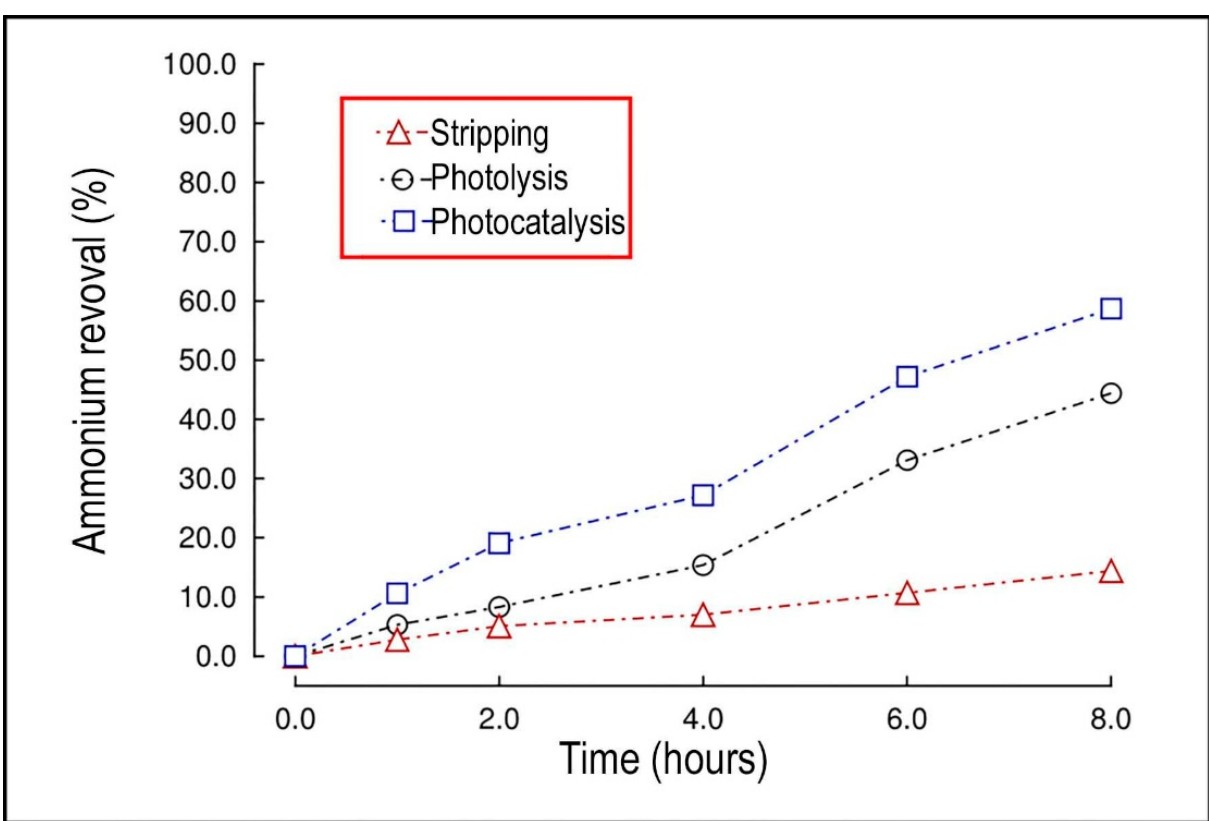

**Figure 5.** Percentage of ammonia removed vs. time by photolysis (o), photocatalysis (□) and stripping (Δ) treatments at pH = 11.0, 25 W UV-C lamp. Q = 1000 L/h, temperature = 20 ± 1 °C.

The time range used in this work and the yields achieved are similar to the processes currently used to remove ammonia nitrogen, such as the annamox process (8-h cycles), responsible for 50% of the nitrogen turnover in marine environments at various temperature and salinity conditions [55].

Considering that the stripping effect is very similar in all the experiments performed at the same recirculation flow rate, it can be considered that the ammonia removal would be carried out by competitive or parallel reactions, following kinetics of order one. Expressing the kinetic equations of the model in an integrated way:

$$C = C_o \, e^{-(k_{PC}+k_{PL})t}$$

$$C = C_o \, e^{-k_{op}.t}$$

where $C$ and $C_o$ are the ammonia concentrations at times t and 0, respectively; $k_{PC}$ is the rate constant of the photocatalytic process and $k_{PL}$ is the rate constant of the photolysis process, the operational constant $k_{op}$ being equal to the sum of both constants. The photocatalytic rate constant ($k_{PC}$) would in turn be an apparent constant that would encompass the reactions of ammonia oxidation to nitrogen gas and nitrate depending on a reaction mechanism that is difficult to elucidate.

Figure 6 shows the operational rate constants ($k_{op}$) for $NH_4^+/NH_3$ removal at different pHs and lamp irradiation powers, following a kinetic curve of monoexponential progress to the baseline. According to the observation, the degradation does not seem to tend to zero, i.e., total elimination of the compound, but to an asymptotic degradation value (C). The results were fitted to a monoexponential model with decrease to a baseline (y = a e$^{-bx}$ + C, where C is the saturation value). This fact has also been observed by Murgia et al. [42].

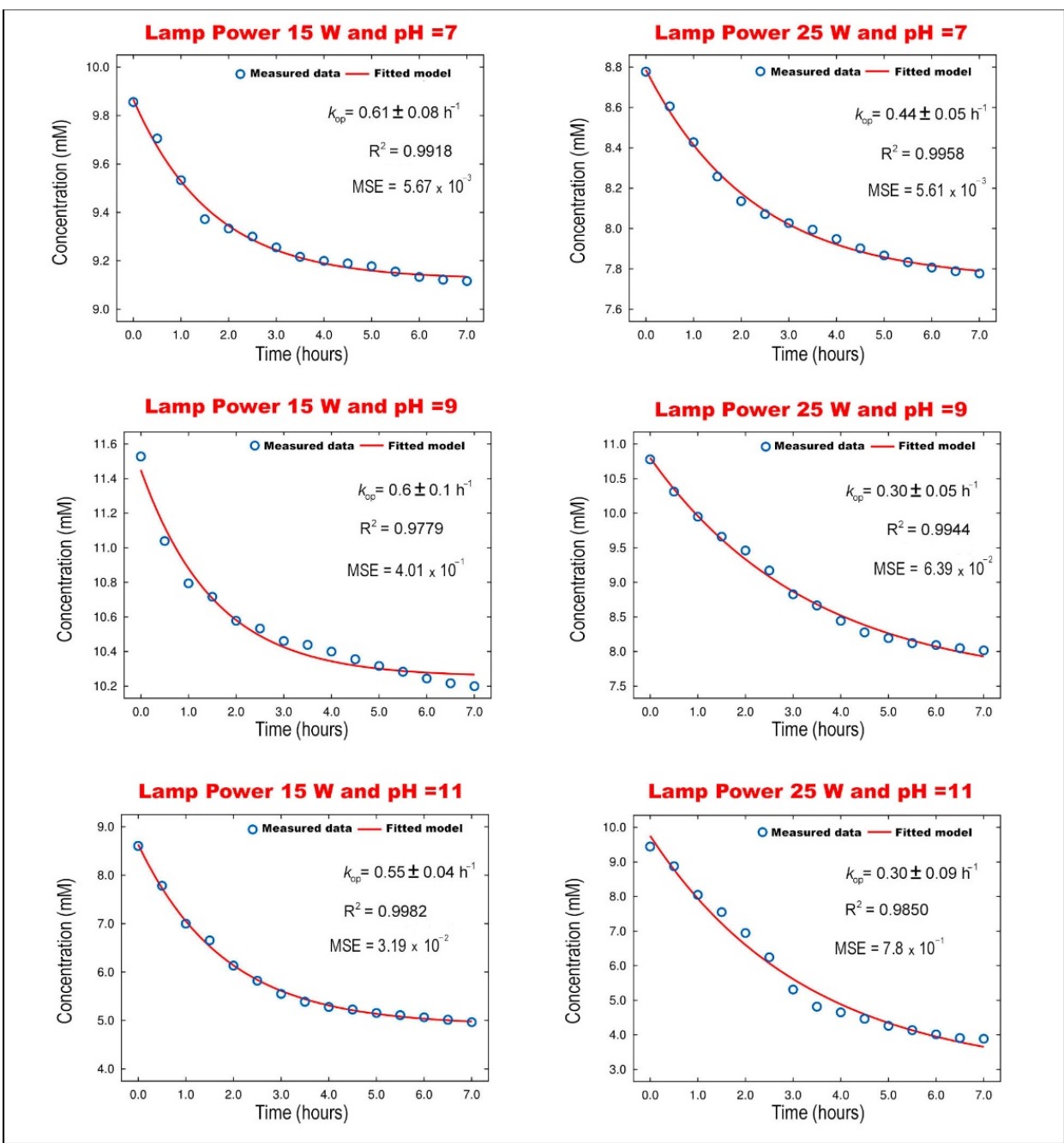

**Figure 6.** Fit of the kinetics of $NH_4^+$/$NH_3$ degradation by photocatalysis to a monoexponential progress kinetic curve to baseline.

The fits are good (coefficient of determination $R^2$ = 0.9779–0.9982, mean squared error MSE = $5.61 \times 10^{-3}$–$7.8 \times 10^{-1}$) but there is no agreement between the degree of $NH_4^+$/$NH_3$ removal and the operational rate constants obtained at different pHs and lamp irradiation powers, since the operational rate constant ($k_{op}$), according to the observed experimental facts, should increase with pH and lamp irradiation power. Thus, from the results obtained, it could be thought that in the $NH_4^+$/$NH_3$ removal, intermediate species formed could either participate in other types of reactions or that different species of the reaction mechanism are involved in different reactions on the surface of the photoreactor or in combination, being adsorbed and saturating the photoreactor surface (asymptotic value).

In this sense, it could also be thought that $NH_4^+$/$NH_3$ species are not adsorbed, intermediate species are, which would explain why the removal values tend towards an

asymptotic value and the degradation rate does not increase, according to the fits regarding pH and irradiation power of the lamp (Figure 6), contrary to what was expected. On the one hand, the increase in $NH_4^+/NH_3$ removal has a strong effect on the ammonia species ($NH_3$) but not so much on the ammonium ion ($NH_4^+$), which could indicate that the two species behave differently on the catalyst surface as asserted by some authors [16,28,32]. On the other hand, the increased conductivity and low formation of nitrite seems to indicate that this species is rapidly formed and decomposed. Nitrite can participate in both the photocatalytic oxidation of nitrite into nitrate [42], and in the photolysis decomposition of intermediate ammonium species ($NH_x$) to nitrogen gas [37]. Finally, it should be noted that, according to the mass balance carried out, it is observed that the formation of nitrogen gas always reaches a maximum, i.e., saturation of the photocatalyst surface occurs, which may be due to intermediate species that saturate the active sites of the fiber surface, causing a decrease in the saturation rate that corresponds to the values of the operational rate constants shown in Figure 6.

It is interesting to note that the experimental data were also fitted (data not shown) according to the model equations for the case of $NH_4^+/NH_3$ removal following a monoexponential progress kinetic curve with a tendency to zero, i.e., total $NH_4^+/NH_3$ elimination. However, in this case the results show a poor fit of this model to the experimental data, except for the case of the largest performance of the reaction at pH 11.0 and 25 W irradiation power of the UV-C lamp, which gives a good fit, which would indicate that this phenomenon of reaction inhibition by adsorption of species on the surface of the photocatalyst has a greater effect the slower the rate of the reaction, i.e., at pH 7.0 and 9.0 and 15 W UV-C lamp irradiation power. Therefore, a study of the proposed mechanism under the most favorable ammonium removal conditions (pH = 11.0 and P = 25 W UVC) is addressed in the following section.

*2.4. Initial Ammonia Removal Rates at pH = 11.0 and 25 W UVC Ultraviolet Lamp Irradiation Power. Models Discrimination*

To discriminate between the possible existence of parallel reactions and the probable intervention of different intermediate species in the reaction mechanism, the kinetics of the degradation of ammonia is also studied by the differential method of the initial rates, measuring these as a function of the initial concentration of the substrate, reducing, under these conditions, to a minimum, the influence of such reactions and the adsorption effects on the photocatalyst surface by reactants and intermediate species to a minimum.

The best fits of the experimental kinetic curves were to the monoexponential models ($R^2 = 0.9756$–$0.9947$, MSE = $6.1 \times 10^{-3}$–$6.0 \times 10^{-2}$) and the corresponding values of the initial rates $v_o$ are shown on Figure 7 for the different ammonia initial concentrations. The green dotted line marks the tangent to the curve whose slope represents $-v_o$ and the black dashed line indicates the asymptotic value.

Different authors [16,18,42] state that the kinetics of photocatalytic ammonium/ammonia removal reactions follow the recommended Langmuir–Hinshelwood model, shown by the differential equation:

$$v_o = -\frac{dC}{dt} = \frac{k\,K\,C_o}{1 + K\,C_o}$$

where $k$ is the reaction rate constant (when $1 << KCo$), K is the ammonium/ammonia adsorption equilibrium constant on the photocatalyst, $C_o$ is the initial ammonium/ammonia concentration. The results of the fit $v_o\,(C_o)$ are shown in Figure 8, where a small curvature of the line fitted to the experimental points can be seen, which together with the statistical quality of the fit ($R^2 = 0.9942$, MSE = $1 \times 10^{-2}$), seems to confirm the trend towards the behavior of the Langmuir–Hinshelwood equation and mechanism in this ammonium/ammonia concentration range.

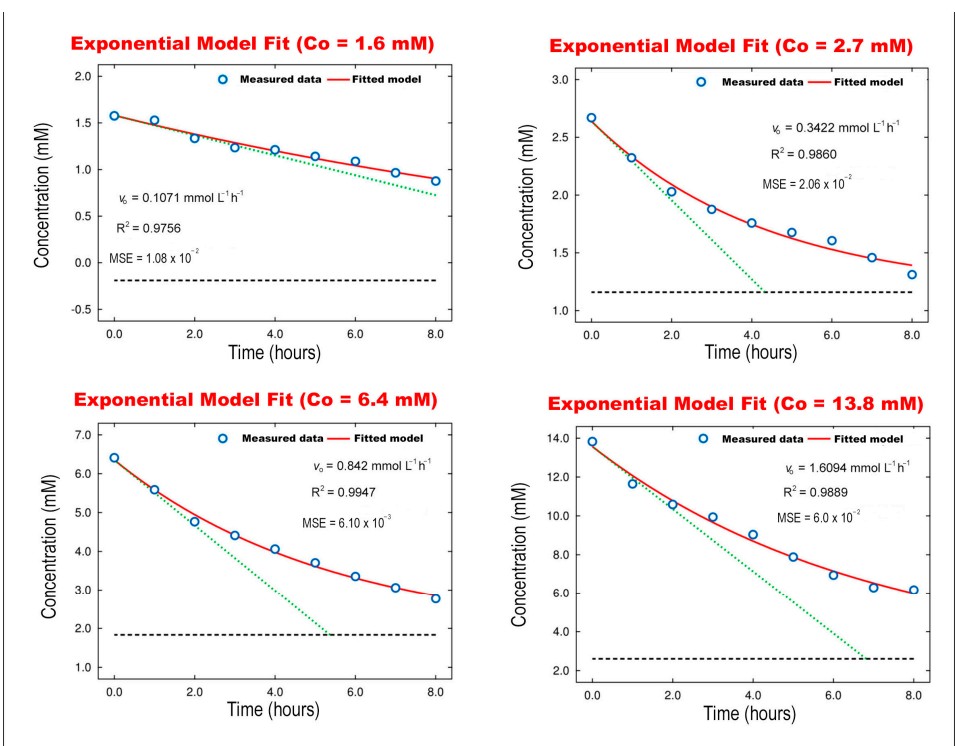

**Figure 7.** Initial ammonia removal rates by photocatalytic process at different initial concentrations of ammonia (at pH = 11.0, P = 25 W, Q = 1000 L/h, temperature = 20 ± 1 °C).

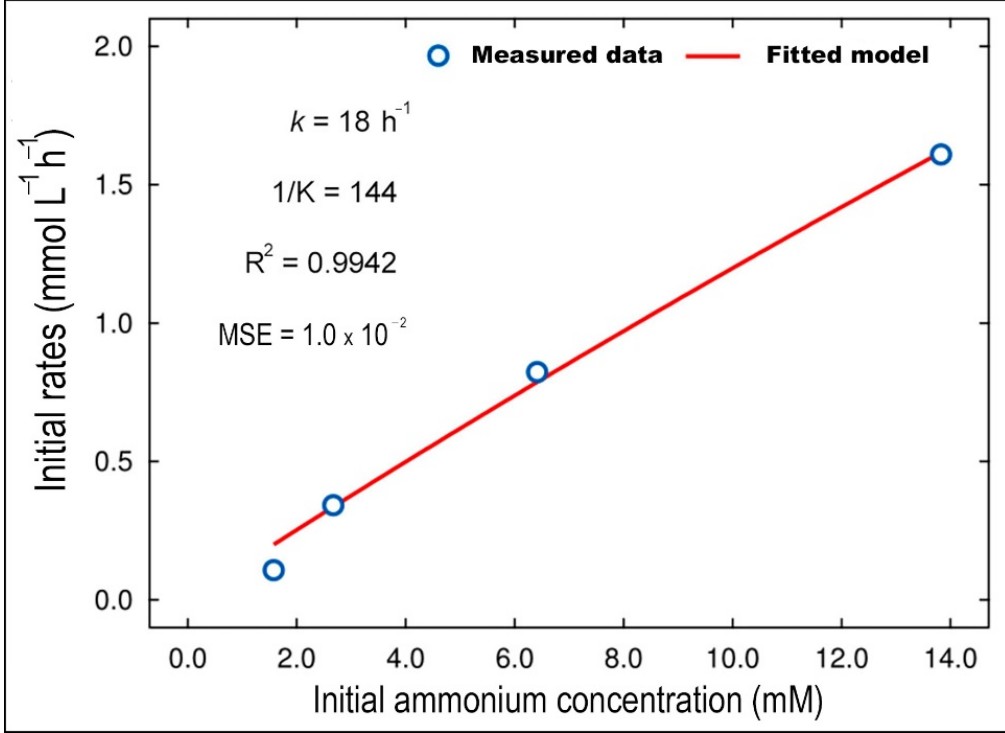

**Figure 8.** Fit to the Langmuir–Hinshelwood differential equation of the initial rate versus initial concentration data for ammonia removal photocatalytic process at pH = 11.0, P = 25 W, Q = 1000 L/h, temperature = 20 ± 1 °C.

Integrating the Langmuir–Hinshelwood differential equation, the corresponding integrated equation can be expressed as:

$$\frac{1}{K} ln\left(\frac{C_t}{C_o}\right) + (C_t - C_o) + kt = 0$$

The constants obtained from Figure 8, the rate constant ($k = 18$ h$^{-1}$) and the adsorption equilibrium constant ($K = 0.0069$), as well as the operational constant defined as the product of both constants ($k_{op} = k\,K = 0.13$ h$^{-1}$), were taken as initial estimates with which we proceeded to close the fitting of the integrated Langmuir–Hinshelwood equation for each of the global ammonium removal kinetic curves shown in Figure 7. This procedure will allow discovering of the sensitivity and accuracy of the fit over time and not only at initial times of ammonium removal reaction, thus indicating if any interference has occurred during the course of the reaction time.

Figure 9 shows the good nonlinear regression fits to the experimental points $C_t\,(t)$ of the integrated rate equation of the Langmuir–Hinshelwood model ($R^2 = 0.9650$–$0.9871$, MSE = $1.1 \times 10^{-2}$–$7.4 \times 10^{-2}$), as well as the values of the parameters, rate constant ($k$) and inverse of the equilibrium adsorption constant ($1/K$). A slight increase in the rate constant of the reaction is observed as a function of the concentration ($k = 12$–$22$ h$^{-1}$), while the value of the adsorption equilibrium constant remains constant ($K = 0.005$) as the ammonia initial concentration increases.

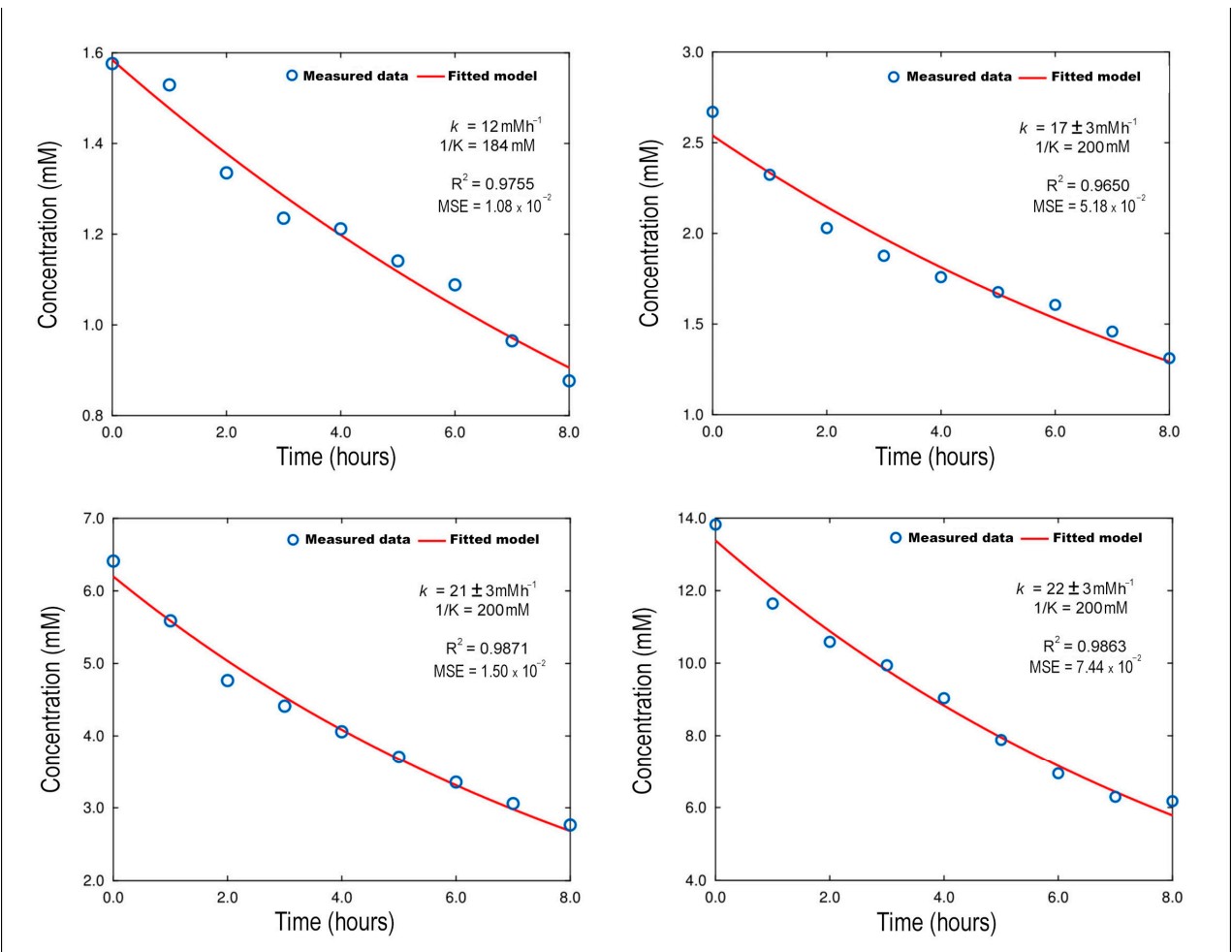

**Figure 9.** Fit to the integrated Langmuir–Hinshelwood rate equation of the experimental data $C_t\,(t)$ of the ammonia removal photocatalytic process at different initial ammonia concentrations (pH = 11.0, P = 25 W, Q = 1000 L/h, temperature = 20 ± 1 °C).

Under the conditions studied, it is observed that the operational rate constant was very low ($0.07$–$0.11$ h$^{-1}$) and no significant variation of the operational rate constant with substrate concentration was observed, remaining practically constant. This indicates a low influence of surface phenomena on the reaction rate, in which the limiting step would be the transfer of matter to the surface of the photocatalytic $TiO_2$-$SiO_2$ glass fiber.

To verify this fact, the surface coverage is determined. The surface coverage $\Theta$ can be related to the substrate concentration $C$ and the apparent adsorption constant at equilibrium $K$, by means of the equation:

$$\Theta = \frac{K\,C}{1 + K\,C}$$

Thus, the value of surface coverage will be between 0 and 1, as it indicates the ratio of occupied sites to total sites, tending to zero when all sites are free and to one when all sites are occupied. Table 1 shows the surface coverage values for the different ammonium initial concentrations:

**Table 1.** Photocatalyst surface coverage as a function of initial ammonia concentrations.

| Initial Concentration (mM) | 1.57 | 2.67 | 6.41 | 13.82 |
|---|---|---|---|---|
| Surface coverage $\Theta$ | 0.008 | 0.013 | 0.032 | 0.07 |

The surface coverage values are very low, not exceeding, at the highest concentration, 10% of the surface coating, which indicates a low adsorption of ammonia species on the surface of the photocatalytic fiber. Likewise, the operational rate constant is very low because the adsorption equilibrium constant is very low as well, which seems to indicate, according to the values obtained from the surface coverage, little adsorption of the substrate on the photocatalyst surface. This could be either because other species or intermediates are adsorbed, or because of the low affinity of this species for the active sites, in this case the homogeneous phase being predominant over the heterogeneous phase. By way of comparison, it is interesting to note that the values of the different parameters of the Langmuir–Hinshelwood equation obtained (Figure 9) are similar to those found in other studies carried out with titanium dioxide in homogeneous aqueous solutions [42]. Other authors indicate discrepancies between the degree of correlation attainable between parameters deduced from such adsorption studies (Langmuir–Hinshelwood model) and those deduced from measurements of relative efficiencies and solute concentration dependence of titanium dioxide ($TiO_2$)-sensitized photocatalyzed degradation of the model pollutants, because this should not be ignored, for poorly adsorbing pollutants, of the roles of solvent molecules at the micro-interfaces since, in reality, polar solvent molecules are likely to compete strongly against solute species for adsorption sites [56].

This assessment also seems to be in line with the high $NH_4^+$/$NH_3$ degradation performance observed by photolysis (44.1%) compared to that observed by photocatalysis (59.7%). Likewise, it seems to indicate that the reaction intermediates in the homogeneous phase react with the ammonia, mostly converting it into nitrogen gas, since the amount of $NO_2^-$ and $NO_3^-$ products formed is very low. It could also explain why the low concentration of nitrite found would favor its transformation into nitrogen gas by photolytic processes [37], which would also favor the low nitrite to nitrate transformation discovered. Even so, photocatalysis is more efficient at removing $NH_4^+$/$NH_3$ than photolysis.

It is possible that the kinetics of ammonia removal actually tends to zero, considering a monoexponential model with a tendency to zero, but it is also possible that other species saturate the photocatalytic fiber, preventing this tendency. This approximation seems to be confirmed by the C values tending to zero observed in the asymptotes (black dashed line) of Figure 7 (initial ammonia removal rates at different initial concentrations). Therefore, the progress kinetic curves $C_{(t)}$ of the four initial rates are fitted now to monoexponential models with zero baseline trend (Figure 10), being the observed parameters consistent with the values of the operational rate constants obtained by fitting the integrated Langmuir–

Hinshelwood equation (Figure 9). This fit indicates that for the conditions set (pH 11.0), ammonium concentration tends to zero but not under experimental conditions of lower reaction rates (pH 7.0 and pH 9.0 and P 15 W). The surface of the photocatalyst can become saturated by other species that are adsorbed by the photocatalytic fiber, and then the amount of ammonia removal would tend towards an asymptotic value.

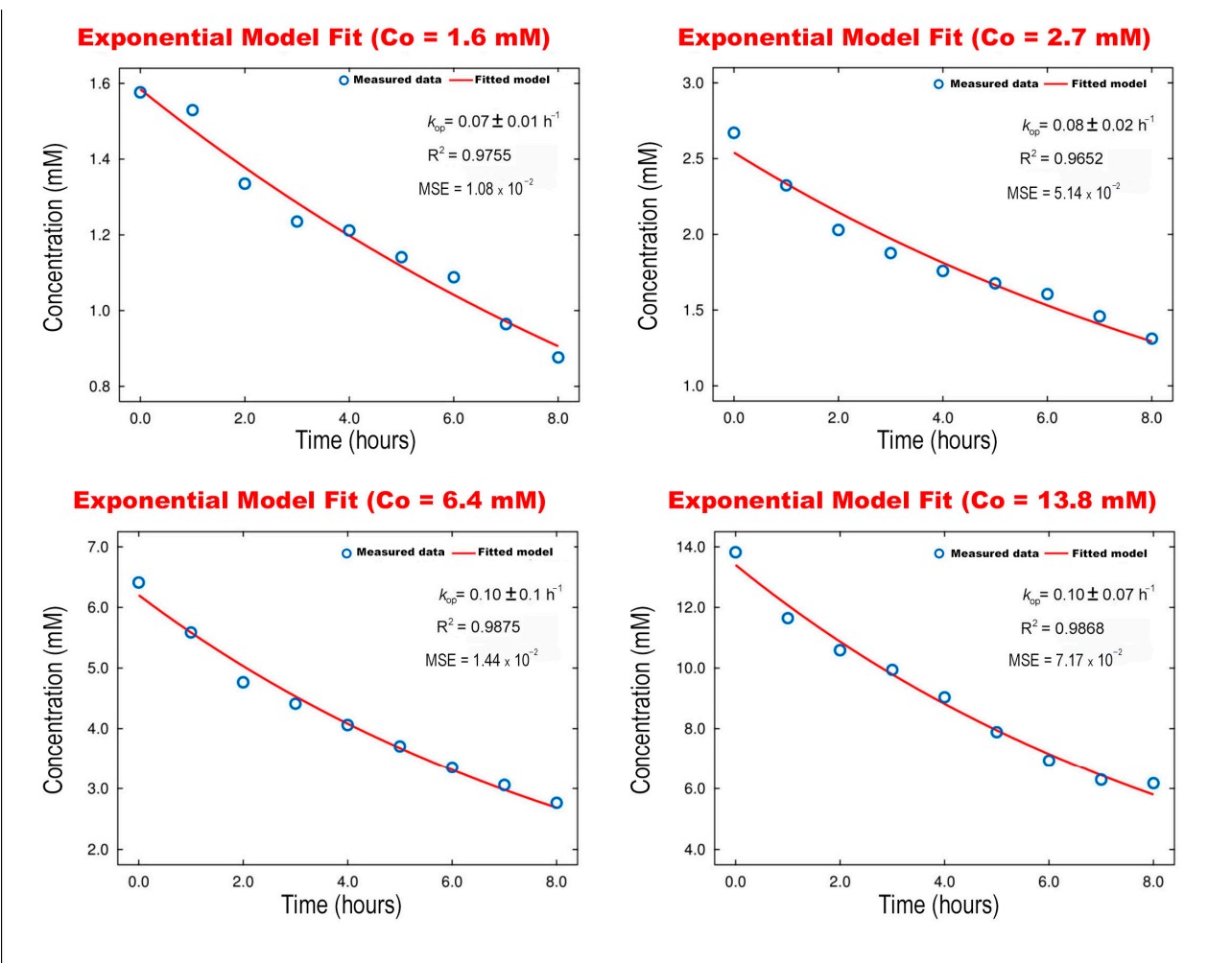

**Figure 10.** Fit to the monoexponential rate equation tending to zero baseline of the experimental points $C_t(t)$ of the ammonia removal photocatalytic process at different initial ammonia concentrations (pH = 11.0, P = 25 W, Q = 1000 L/h, temperature = 20 ± 1 °C).

Finally, for comparative purposes with regard to the photocatalytic process, the photolysis process of ammonia removal is presented under the same optimal experimental conditions, that is, pH = 11.0 and P = 25 W but without photocatalytic fiber. Figure 11 shows the fit of the experimental data $C(t)$ to the integrated rate equation of order 1, being $k_{PL}$, the rate constant of the photolysis process:

$$C = C_o\, e^{-k_{PL}.t}$$

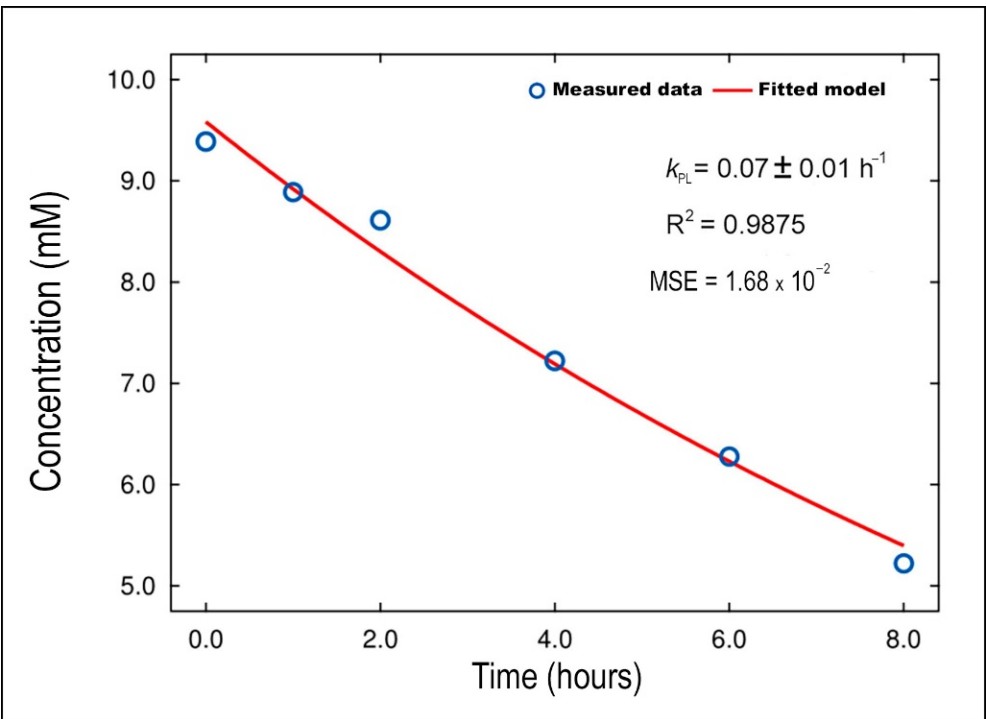

**Figure 11.** Fit of the experimental data $C\,(t)$ of the ammonia removal photolysis process to the first order monoexponential rate equation tending to zero (pH = 11.0, P = 25 W, Q = 1000 L/h, temperature = 20 $\pm$ 1 °C.

As can be seen in Figure 11, the value of the photolysis rate constant ($k_{PL}$) = 0.07 $\pm$ 0.01 h$^{-1}$ is of a similar order of magnitude to the operational photocatalytic rate constant ($k_{op}$) of Figure 10, being greater the photocatalytic effect the higher the initial concentration of ammonia.

### 2.5. Study of the Photocatalytic Fiber Stability after Ammonium/Ammonia Removal Process

A study of photocatalytic fiber degradability was carried out after the $NH_4^+$/$NH_3$ degradation experiments. For this, a structural characterization study was performed, semi-quantitatively identifying, at the surface level, both the chemical elements deposited and those that form the photocatalytic fiber, as well as carrying out 50 and 200 μm scanning electron micrographs (Figure 12).

Photocatalytic fiber at pHs higher than 11.0 deteriorates by losing mass, mainly in the form of silicon, thus no studies of $NH_4^+$/$NH_3$ degradation are carried out at pH greater than 11.0. The results of Figure 12 show a coating of the fiber with impurities of chlorine (from the reagent ammonium chloride), iron, potassium and sodium deposited mainly on the titanium particles, which shows adsorption on the surface of the photocatalyst of species other than ammonium, supporting the empirical results presented above.

### 2.6. A Two Parallel Reaction Mechanism for Ammonium Degradation by Photolysis and Photocatalysis

According to previous studies performed by other authors [14,36–39,42] and the empirical results shown in this work, the degradation of $NH_4^+$/$NH_3$ to two final reaction products, nitrogen gas and nitrate, is carried out through two parallel reactions by simultaneous photolysis and photocatalytic actions, proposing the mechanism of Figure 13, where nitrogen gas is formed by photocatalysis and photolysis processes and nitrate is formed via photocatalysis.

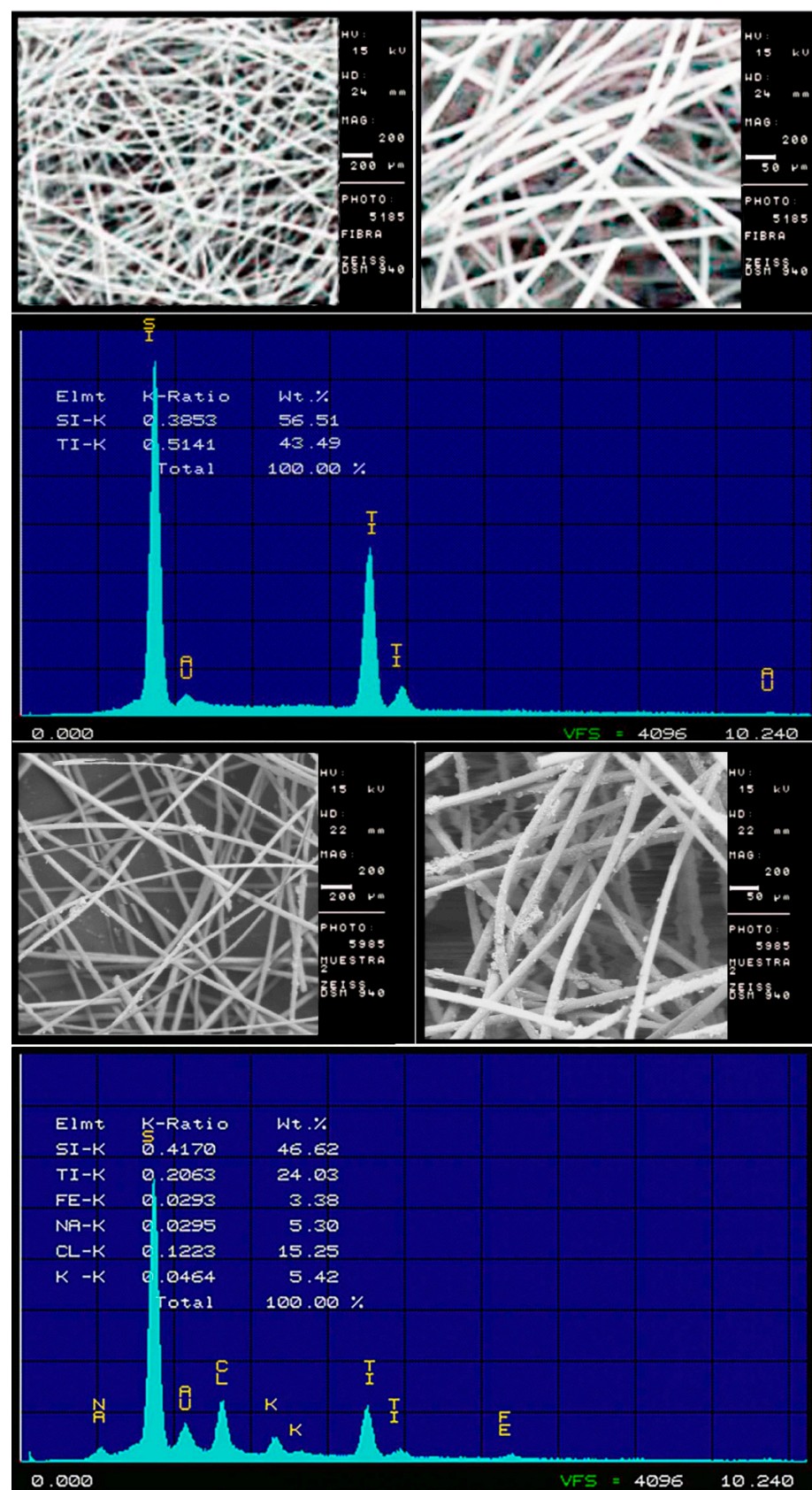

**Figure 12.** Study of the photocatalytic fiber stability after $NH_4^+/NH_3$ removal process: 200 and 50 μm scanning electron micrographs and energy dispersive X-ray elemental microanalysis (EDXMA). Upper panel: fiber virgin and lower panel: fiber after $NH_4^+/NH_3$ removal.

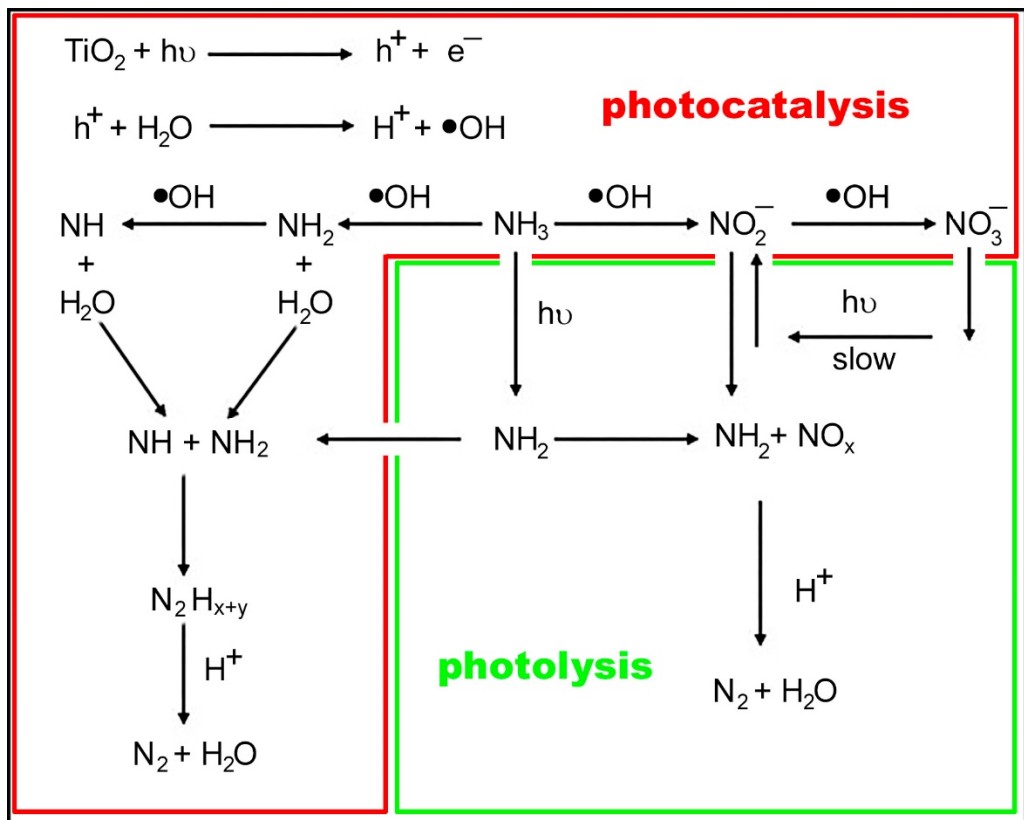

**Figure 13.** The proposed two parallel reaction mechanism for $NH_4^+/NH_3$ degradation by photolysis and photocatalysis.

In order to perform the validation of the proposed model of parallel reactions, a final confirmation of the model consisted of simultaneously carrying out the fits of the respective $C(t)$ data to the differential rate equations for the ammonium degradation and the formation of nitrate and nitrogen gas. Intermediate species, such as nitrite, have not been considered, as their formation and disappearance are very fast and the measured experimental values were found to be too low. According to the scheme presented in Figure 13, their system of differential rate equations would be:

$$\text{v} = -\frac{d[Ammonium]}{dt} = k_1[Ammonium] + k_2[Ammonium] = (k_1 + k_2)[Ammonium]$$

$$\text{v} = \frac{d[nitrate]}{dt} = k_1[Ammonium]$$

$$\text{v} = \frac{d[nitrogen\ gas]}{dt} = k_2[Ammonium]$$

where $k_1$ corresponds to the nitrate formation rate constant and $k_2$ to the nitrogen gas formation rate constant (both by photolysis and photocatalysis) and $k_3$ expressed as the sum of the nitrogen gas and nitrate formation rates ($k_1 + k_2$) would correspond to the ammonium/ammonia degradation overall rate constant ($k_3$). The results of the simultaneous fits of $C(t)$ data to the three differential equations are shown in Figure 14, the blue line showing the fit of the ammonium/ammonia degradation data, the red line, the fit of nitrate formation data and the green line, the fit of nitrogen gas formation data to the corresponding differential rate equations. Table 2 shows the values of $k_1$, $k_2$ and $k_3$ obtained from such fittings.

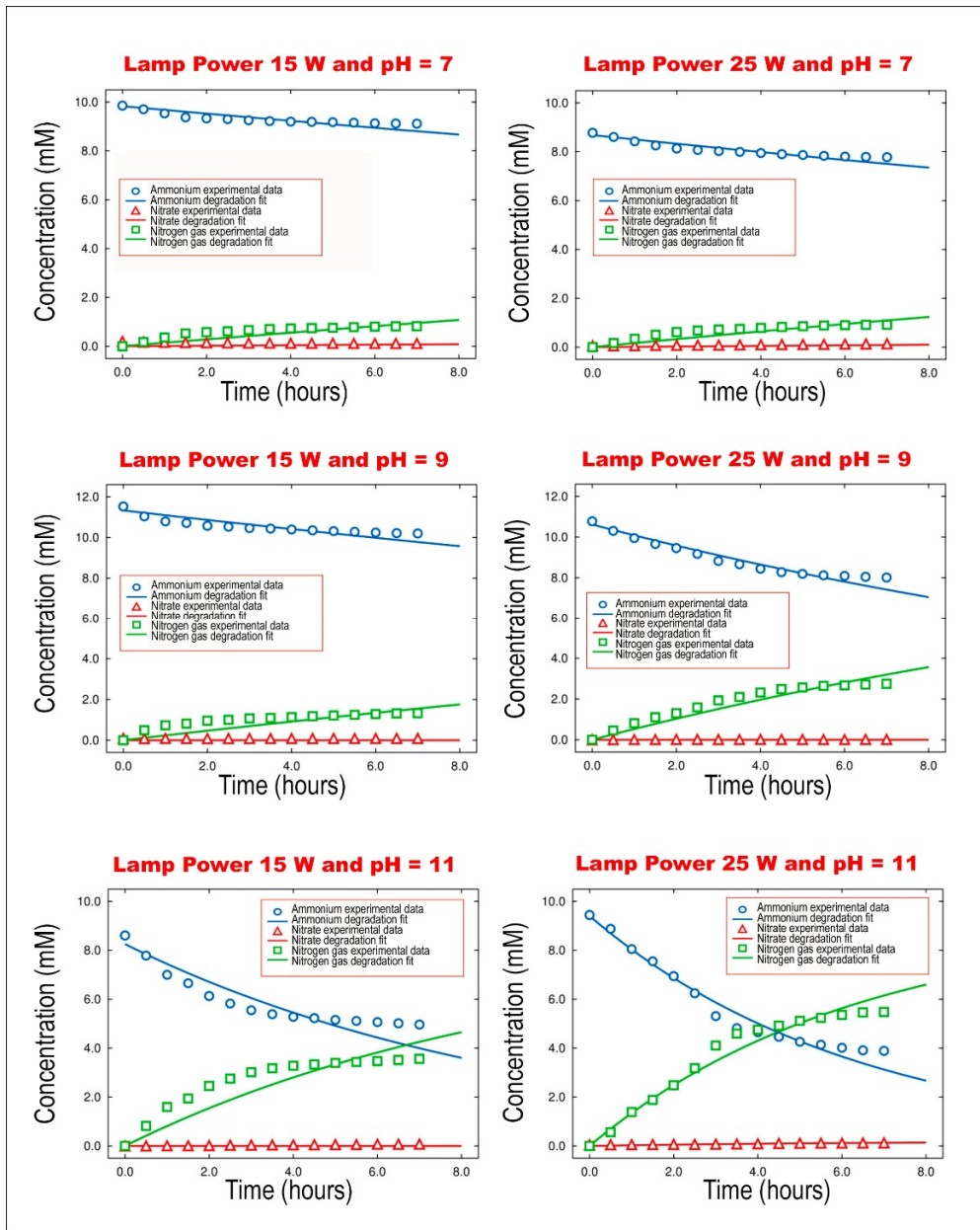

**Figure 14.** Simultaneous fits of 3 sets of $C(t)$ data to the corresponding 3 differential rate equations according to the proposed two parallel reaction mechanism for $NH_4^+/NH_3$ degradation by photolysis and photocatalysis, at two lamp irradiation powers and different pHs, Q = 1000 L/h, temperature = 20 $\pm$ 1 °C.

**Table 2.** Values of the rate constants for the disappearance of ammonium/ammonia ($k_3$) and the appearance of nitrate ($k_1$) and nitrogen gas ($k_2$), obtained by simultaneous fits of the three differential equations proposed to the corresponding sets of experimental $C(t)$ data.

|  | $k_1$ (h$^{-1}$) | $k_2$ (h$^{-1}$) | $k_3$ (h$^{-1}$) |
|---|---|---|---|
| Photocatalysis 15 W pH = 7,0 | $(1.1 \pm 0.2)\ 10^{-3}$ | $(1.5 \pm 0.2)\ 10^{-2}$ | 0.016 |
| Photocatalysis 15 W pH = 9.0 | $(8.2 \pm 0.6)\ 10^{-6}$ | $(2.1 \pm 0.2)\ 10^{-2}$ | 0.021 |
| Photocatalysis 15 W pH = 11.0 | $(2.2 \pm 0.1)\ 10^{-5}$ | $(1.03 \pm 0.09)\ 10^{-1}$ | 0.103 |
| Photocatalysis 25 W pH = 7.0 | $(1.5 \pm 0.2)\ 10^{-3}$ | $(1.9 \pm 0.5)\ 10^{-2}$ | 0.021 |
| Photocatalysis 25 W pH = 9.0 | $(8.7 \pm 0.6)\ 10^{-7}$ | $(5.1 \pm 0.3)\ 10^{-2}$ | 0.051 |
| Photocatalysis 25 W pH = 11.0 | $(3.3 \pm 0.2)\ 10^{-5}$ | $(1.54 \pm 0.07)\ 10^{-1}$ | 0.154 |

The simultaneous fit of the two parallel reactions model to the experimental data was good (in Student´s *t*-test, "*p*-values" of the parameter estimates were less than 0.05), which confirms that the proposed model (Figure 13) fits well the experimental results. It is also observed that the degradation of $NH_4^+/NH_3$ into nitrogen gas is faster than the degradation of $NH_4^+/NH_3$ into nitrate, as would be expected from the observation of the progress curves of Figure 1. Furthermore, the values of the $NH_4^+/NH_3$ degradation rate constants ($k_3$) are in agreement with the values obtained for the operational rate constant from the fit to the integrated Langmuir–Hinshelwood equation at different initial ammonia concentrations (Figure 9) and the values of the operational rate constants obtained by fitting the experimental data to the monoexponential models with zero baseline (Figure 10). Finally, the values of the $NH_4^+/NH_3$ degradation constant ($k_3$) at the different pHs and lamp irradiation powers studied are consistent with the empirical values obtained for the $NH_4^+/NH_3$ removal (Figure 4), with its removal performance being higher at a higher reaction rate. All these evidences seem to confirm that the mechanism of two parallel equations fits the experimental $C(t)$ data for $NH_4^+/NH_3$ degradation by photolysis and heterogeneous photocatalysis.

## 3. Materials and Methods

### 3.1. TiO₂/SiO₂ Fixed Bed Photoreactor with Total Recirculation

The $TiO_2/SiO_2$ fixed bed photoreactor (UBE Industries, Japan) used is shown in Figure 15. The system has a tank for the sample with a capacity of 200 L, a 1 hp pump for recirculation of the sample through the system and, at the outlet of the pump, the water first goes through a 50 µm solid filter and then through a rotameter to measure the sample flow entering the photoreactor body.

The photoreactor is characterized by maintaining a vertical piston flow with bottom inlet and upper lateral outlet; the reactor body is made of stainless steel and has a manometer on the bottom and another on the top. The internal walls of the photoreactor are polished, in such way that the radiation that reaches them is reflected, thus generating greater incidence of light on the photocatalyst.

For the photocatalysis experiments, in addition to the pilot photoreactor described, a $TiO_2/SiO_2$ photocatalyst (UBE Industries, Japan) was used. The semiconductor material used as a non-woven photocatalytic fiber with gradient in the crystalline structure, whose patent belongs to UBE Chemical Industries [22,57], consists of a $SiO_2$ fiber mesh, which supports the $TiO_2$ catalyst, generating a $TiO_2/SiO_2$ catalyst/support system, avoiding the phenomenon of dragging of the photocatalyst from the surface of the support (a phenomenon known as peeling), as a consequence of its friction with the fluid of the liquid. The maximum pressure that the fiber can withstand is up to 10 kg/cm², with optimum performance in the range of 3–6 kg/cm². A $TiO_2$ semiconductor supported on a $SiO_2$ fiber, contained in 4 stainless steel conical meshes placed longitudinally on rods to immobilize them. The photocatalyst is located between the radiation source and the walls of the photoreactor [58]. Two low-pressure mercury lamps were used as irradiation sources: the first one, a 40 W lamp (Philips TUV 36T5 HE 4P SE UNP/32), emitting 15 W of UV-C ultraviolet radiation and the second one, a 75 W lamp (Philips TUV 36T5 HO 4P SE UNP/32), emitting 25 W of UV-C ultraviolet radiation, both with a maximum wavelength of 253.7 nm. Each lamp was placed inside a transparent quartz tube to prevent it from coming into contact with the sample.

### 3.2. Experimental Conditions

In order to establish the experimental conditions for the study of $NH_4^+/NH_3$ degradation in the UV-C photocatalytic reactor, the following considerations were taken into account.

Synthetic wastewater with $NH_4^+/NH_3$ concentrations similar to the outflows of leachate treatment from wastewater treatment plants was used [59] because its removal capacity is limited with $NH_3$-N > 100 mg/L wastewaters [60]. In addition, in order to study the different nitrogen species involved in the $NH_4^+/NH_3$ degradation mechanisms, possi-

ble inhibitions that may be produced by the adsorption of other species on the photocatalyst surface should be avoided [27,33].

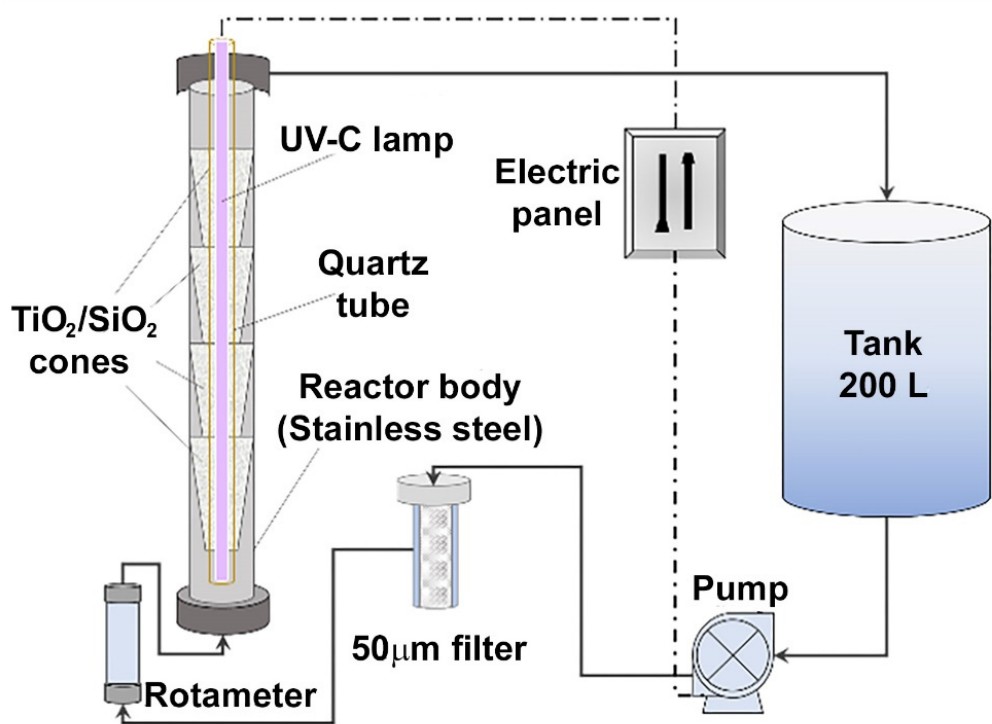

**Figure 15.** Photoreactor system used for photolysis and photocatalytic organic compounds degradation processes.

The experiments with synthetic wastewaters were carried out at pH 7.0, 9.0 and 11.0, proving that pH was an essential factor in the photolytic and photocatalytic degradation of $NH_4^+/NH_3$.

Another important operational parameter is the power of the UV-C ultraviolet irradiation and, therefore, the degradation experiments were carried out with two different lamps (15 W, 25 W), since the rate of photocatalytic activation and the formation of the electron-hole pair are strongly influenced by the power of UV-C irradiant light [34].

All experiments were carried out with 50 L of wastewater synthetic samples with initial ammonium concentrations (ammonium chloride provided by Sigma Aldrich) in the range 180–200 mg/L in the feed tank of the UV photoreactor. In the photocatalysis experiments, 4 cones of $TiO_2/SiO_2$ photocatalyst were used. For the photolysis experiments, the cones were taken out of the photoreactor. The pump is then switched on, adjusting the flow rate to 1000 L/h, maintaining a constant temperature at $20 \pm 1\,°C$, by means of a refrigerant cooling coil, adjusting and maintaining constant pH (7.0, 9.0 and 11.0) with an 8% (m/v) NaOH aqueous solution and carrying out the experiments for 7–8 h under UV-C light. For each sample, in addition to temperature and pH, conductivity, dissolved oxygen, ammonium/ammonia, nitrite and nitrate concentration were measured by selective electrodes of the YSI6920 multiparameter probe. In the ammonium measurements throughout this work, the sum of the 2 conjugated $NH_4^+/NH_3$ species are expressed. Similarly, nitrogen species were determined by electronic absorption spectroscopy using a PG T80+ spectrophotometer and an HACH DR/2010 photometer. The stability studies of the photocatalytic fiber were carried out by energy dispersive X-ray elemental microanalysis (EDXMA), a technique which was also used to measure the titanium/silicon ratio of the fiber as well as the percentage of titanium on the surface fiber.

*3.3. Data Analysis*

Free open-source software was used for the different studies. For the chemical kinetics studies of the $NH_4^+/NH_3$ degradation tests, modelling and validation of the proposed mechanism, the statistical package SIMFIT was used. This package was developed at the University of Manchester by William G. Bardsley (http://www.simfit.org.uk) (accessed on 14 December 2021). The Spanish version of SIMFIT is maintained by F. J. Burguillo of the University of Salamanca (http://simfit.usal.es) (accessed on 16 December 2021) [61].

The "INRATE" program was used for the calculation of the initial velocities at the different substrate concentrations. The Langmuir–Hinshelwood integrated rate equation was fitted to the appropriate model using the "QNFIT" program and the differential equations were simultaneously fitted to the experimental data using the DQSOL program of the SIMFIT statistical package.

**4. Conclusions**

Under the conditions used, $NH_4^+/NH_3$ can be decomposed both by photolysis (UV radiation action) and photocatalysis (UV radiation plus photocatalytic fiber) routes, without neglecting the volatilization of ammonia by stripping processes (agitation and volatilization). It was found that the $NH_4^+/NH_3$ removal percentage is mainly a function of pH and lamp irradiation power, the higher the $NH_3/NH_{4+}$ removal performance the higher the lamp irradiation power and the more basic the pH. An analysis of the nitrogen species occurring during the $NH_4^+/NH_3$ degradation process revealed that it decomposes mainly into nitrogen gas and nitrate, the intermediate species nitrite being very unstable, as it is rapidly formed and transformed into other species. Dissolved oxygen does not have a great influence on the reaction and remains almost constant. On the other hand, the conductivity increases with pH and lamp irradiation power, indicating that there is a greater formation of ionic intermediate species. The experimental data fit well to a Langmuir–Hinshelwood adsorption model. The low adsorption equilibrium constants (K = 0.0069) and the surface coverage values that for high $NH_4^+/NH_3$ concentrations do not exceed 10% coverage, show a low affinity of ammonium/ammonia for adsorption and surface reaction on the photocatalytic fiber, which translates into low operational rate constants ($k_{op} = 0.13$ $h^{-1}$). The photolysis rate constant being $k_{PL} = 0.07 \pm 0.01$ $h^{-1}$.

The kinetic progress curves for $NH_4^+/NH_3$ degradation reaction tend towards plateau-type asymptotic values and not towards total $NH_4^+/NH_3$ removal, thus it could be thought that the reaction is inhibited by the adsorption of intermediate species on the surface of the photocatalytic fiber. By studying the initial reaction rates at different initial ammonia concentrations, it was shown that the decomposition of ammonia should tend to zero, or total removal, but the adsorption of intermediate species on the photocatalytic fiber, especially at low reaction rates (less basic pHs and lower lamp irradiation power) causes saturation of the fiber and explains the maximum limit of degradation. This fact was corroborated by the analytical semi-quantitative study (EDXMA) and the structural characterization by scanning electron microscopy of the photocatalytic fiber coating, which seems to confirm the hypothesis of adsorption of intermediate species on the photocatalyst surface.

The good fits of the experimental $C(t)$ data to a model of two parallel $NH_4^+/NH_3$ decomposition reactions confirm the proposed $NH_4^+/NH_3$ degradation mechanism, which consists, on the one hand, in the formation of nitrogen gas and, on the other hand, in the formation of nitrate. At the optimal conditions, the rate constants for the disappearance of ammonia were $k_3 = 0.154$ $h^{-1}$ and for the appearance of nitrate and nitrogen gas $k_1 = 3.3 \pm 0.2 \ 10^{-5}$ $h^{-1}$ and $k_2 = 1.54 \pm 0.07 \ 10^{-1}$ $h^{-1}$, respectively.

This UV-C photocatalytic process is shown to be more effective for degrading $NH_4^+/NH_3$ than the photolytic process and does not require the addition of reagents, such as $H_2O_2$, for the formation of hydroxyl radicals and could compete with the processes carried out by adapted anammox bacteria in an SBR bioreactor. This comparative study will be the next target of future research.

**Author Contributions:** Methodology, J.C.G.-P. and M.G.-R.; validation, J.C.G.-P. and M.G.-R.; formal analysis, J.C.G.-P. and M.G.-R.; investigation, J.C.G.-P. and M.G.-R.; resources, J.C.G.-P., M.G.-R. and J.B.P.-N.; data curation, L.A.G.-B., M.G.-R. and J.C.G.-P.; writing—original draft preparation, J.C.G.-P. and M.G.-R.; writing—review and editing, J.C.G.-P., M.G.-R. and J.B.P.-N.; visualization, J.C.G.-P. and M.G.-R.; supervision, J.C.G.-P., L.A.G.-B., M.G.-R. and J.B.P.-N.; project administration, M.G.-R.; funding acquisition, J.C.G.-P. and M.G.-R. All authors have read and agreed to the published version of the manuscript.

**Funding:** This research was funded by the Consejo Nacional de Ciencia y Tecnología (CONACyT), Instituto Politécnico Nacional (IPN/SIP project 20190247and 20200670). This research also received the financial support and the supply of the photocatalytic fibre pilot plant from UBE Corporation Europe S.A. The content does not necessarily reflect the views and policies of the funding organizations.

**Institutional Review Board Statement:** Not applicable.

**Informed Consent Statement:** Not applicable.

**Data Availability Statement:** Data are contained within the article.

**Acknowledgments:** The authors wish to thank UBE Corporation Europe S.A. for supplying the photocatalytic fiber pilot plant and financially supporting this project. L.A.G.-B. and J.B.P.-N. thank the Consejo Nacional de Ciencia y Tecnología (CONACyT), who provided funding.

**Conflicts of Interest:** The authors declare no conflict of interest.

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
