# Peer review of "Kinetic Study and Modeling of the Degradation of Aqueous Ammonium/Ammonia Solutions by Heterogeneous Photocatalysis with TiO2 in a UV-C Pilot Photoreactor"

_catalysts, doi:10.3390/catal12030352_

Round 1

Reviewer 1 Report

The authors presents kinetic study and modeling of the degradation of aqueous ammonium/ammonia solutions by heterogeneous photocatalysis with TiO2 in a UV-C pilot photoreactor. . My comments are the following:

  1. In line 94, the reaction pathway is not complete. The N in the end should be N atom adsorbed on the electrode surface and this reaction chain finish by the recombination of two N atom with N2 gas production.  
  2. In fig.1, it is almost impossible to follow the production of nitrate from the figure. I suggest to use a new layer (Right Y) for nitrate concentration.
  3. Figure 4 indicate that the degree of NH4+/NH3 removal increases with pH at both lamp irradiation powers. Why not choose a higher pH (>11).
  4. The scales are missing in the SEM images Fig.12.  Three broken lines are observed in the right SEM images and the quality of this image could be improved by adjusting the scan speed.
  5. Line 413, Figure 13 should be Figure 12.
  6. To support the stability study, a controlled trial should be done by comparing the SEM-EDX result with photocatalytic fibers before and after  photocatalytic study in order to show the morphology and elementary composition stability .

Reviewer 2 Report

The authors have revisited the transformatios of NH4+/NH3 in aqueous solution at various pH (7-11 range) under UV-C irradiation with or without TiO2 immobilized on a SiO2 fiber mesh. They have discriminated the effects due to photocatalytic, photolytic and stripping phenomena.

Please find below some recommendations to improve the manuscript.

Format

1. I think the manuscript is long with respect to its content. The writing appears to me often circuitous; it could be more effective. I recommend the authors trying to shorten it; additionally, are all the figures really needed?. Perhaps, the authors might ask a colleague, who was not involved in this study, to read the manuscript and make suggestions.

2. Personally (and I think most readers as well), I would like to know the experimental conditions before the presentation of the results. Therefore, I suggest reversing sections 2 and 3 unless the Journal imposes this organization.

Content

1. The authors must add “under the conditions used” in both the abstract and the conclusions to emphasize that the values indicated are relative and not absolute.

2. The amounts of N2 formed were calculated. The authors should indicate whether they try to detect N2 and what was the detection limit.

3. In the same line, they should indicate whether they try to detect nitrogen oxides in the gas phase.

4. Discrepancies between adsorption measurements and adsorption coefficients deduced from the Langmuir-Hinshelwood model were reported previously. I think the authors must indicate that and cite, for instance, this quite informative article: Cunningham, J., Al-Sayyed, G., Srijaranai, S. Adsorption of model pollutants on TiO2 particles in relation to photoremediation of contaminated water In Helz, G., Zepp, R., & Crosby, D. (1995). Aquatic and Surface Chemistry. Lewis, Boca Raton, FL, 1994, pp. 317-348.

5. Lines 62-75. In this paragraph, the authors could delete the line mentioning metal deposits on TiO2 since the material they used did not contain such deposits. By contrast, I suggest they mention a paper (Mozzanega, H., Herrmann, J. M., & Pichat, P. (1979). Ammonia oxidation over UV-irradiated titanium dioxide at room temperature. Journal of Physical Chemistry, 83 (17), 2251-2255) about the photocatalytic oxidation of ammonia in the gas phase because it concerns their target pollutant.

6. Equations from line 94 to 120 and Fig 14. Please rewrite so that the overall charge is equal on both sides of each equation.

7. Labeling error: fig 4/5. Please correct.

Round 2

Reviewer 1 Report

The manuscript could be accepted.